# Synthesis of Fe- and Co-Doped TiO_2_ with Improved Photocatalytic Activity Under Visible Irradiation Toward Carbamazepine Degradation

**DOI:** 10.3390/ma12233874

**Published:** 2019-11-24

**Authors:** Abderrahim El Mragui, Yuliya Logvina, Luís Pinto da Silva, Omar Zegaoui, Joaquim C.G. Esteves da Silva

**Affiliations:** 1Chemistry Research Unit (CIQUP), Faculty of Sciences of University of Porto, Rua do Campo Alegre 697, 4169-007 Porto, Portugal; a.elmragui@edu.umi.ac.ma (A.E.M.); up201407073@fc.up.pt (Y.L.); luis.silva@fc.up.pt (L.P.d.S.); 2Research team “Materials and Applied Catalysis: MCA”, “CBAE” Laboratory, Faculty of Sciences, Moulay Ismail University, BP.11201 Zitoune, Meknès, Morocco; o.zegaoui@fs.umi.ac.ma; 3LACOMEPHI, GreenUPorto, Department of Geosciences, Environment and Territorial Planning, Faculty of Sciences, University of Porto, Rua do Campo Alegre 697, 4169-007 Porto, Portugal

**Keywords:** doped TiO_2_, photocatalytic degradation, pharmaceuticals, carbamazepine, UV-A light, visible light

## Abstract

Pure TiO_2_ and Fe- and Co-doped TiO_2_ nanoparticles (NPs) as photocatalysts were synthesized using wet chemical methods (sol-gel + precipitation). Their crystalline structure and optical properties were analyzed using X-ray diffraction (XRD), Raman spectroscopy and Fourier-transform infrared (FTIR) spectroscopy, ultraviolet-visible light (UV-Vis) diffuse reflectance spectroscopy (DRS), and photoluminescence (PL) spectroscopy. The photocatalytic activity of the synthesized nanoparticles was evaluated through degradation of carbamazepine (CBZ) under UV-A and visible-light irradiations. The XRD and Raman analyses revealed that all synthesized nanomaterials showed only the anatase phase. The DRS results showed that the absorption edge was blue-shifted for Fe-doped TiO_2_ NPs. The decrease in charge recombination was evidenced from the PL investigation for both Co-doped and Fe-doped TiO_2_ nanomaterials. An enhancement in photocatalytic degradation of carbamazepine in aqueous suspension under both UV-A light and visible-light irradiations was observed for Fe-doped Titania NPs by comparison with pure TiO_2_. These results suggest that the doping cations could suppress the electron/hole recombination. Therefore, the photocatalytic activity of TiO_2_-based nanomaterials was enhanced.

## 1. Introduction

The contamination of water systems by pharmaceuticals is recognized as an environmental issue. Those pharmaceutical compounds are discharged from private households and hospitals and reach wastewater treatment plants. Many of these compounds cannot be fully removed by these plants and they are discharged in surface water, raising concerns about their potential ecotoxicological effects [1,2,3]. One of the pharmaceuticals frequently detected in the aquatic environment is carbamazepine (CBZ), also known as Tegratol and Epitol, which is a prescription drug used extensively in the clinical treatment of epilepsy, trigeminal neuralgia, and other psychiatric disorders [1,4,5,6]. Carbamazepine was detected in the environment with concentrations ranging from ng/L up to µg/L [1,7,8]. Despite its low concentration in the environment, CBZ is resistant to biodegradation and to conventional wastewater treatment processes, and it is retained in the environment for a long time. Given these facts, CBZ is classified as a persistent organic pollutant [9,10,11]. Its accumulation poses a threat to the quality of water resources, and it is suspected to pose a toxic effect on aquatic organisms [3].

Advanced oxidation processes (AOPs) were proven to be effective for the treatment of wastewater [1,12]. Among the various AOPs, heterogeneous photocatalysis is selected as one of the best options for the destruction of many recalcitrant organic pollutants including CBZ [12,13]. One of the most important aspects of heterogeneous photocatalysis is the selection of the photocatalyst. Currently, pure or modified titanium-dioxide nanoparticles are the most extensively used photocatalysts due to the optical and electronic properties of Titania, and their low cost, abundance, chemical stability, and non-toxicity [12,14]. Unfortunately, the photocatalytic efficiency of TiO_2_ is still not satisfactory under visible-light irradiation because of its wide band gap (E_g_ ≈ 3.20 eV) [15,16] that allows it to absorb only ultraviolet (UV) wavelengths, making TiO_2_ useful for wastewater treatment only in the ultraviolet range of sunlight, which greatly inhibits its industrial application [16,17,18,19,20]. Therefore, several studies focused on the development of visible-responsive TiO_2_ photocatalysts either by doping with metal ions and/or non-metal ions or by creation of hetero-junctions with other semiconductors [18,21,22]. Doping with transition-metal ions, such as Fe, Co, Mn, and Ni ions, was reported to be effective for enhancing the photocatalytic activity of TiO_2_ [22,23,24]. In fact in our previous studies, which included the effect of Mn, Co, and Ni mono-doping on TiO_2_ nanoparticles and the effect of co-doping with metal and non-metal ions, we reported that the mono-doping of TiO_2_ with Co and its co-doping with (P, Mo), (P, W), or (Si, W) improved the photocatalytic activity of TiO_2_ under visible-light irradiation, toward methyl orange, in comparison with undoped TiO_2_ [25,26].

Therefore, in order to continue our studies about the effect of doping with transition-metal ions on the structural, optical, and photocatalytic properties of TiO_2_ nanoparticles, the present study examines the effect of Fe doping on TiO_2_ nanoparticles, compared to Co doping and pure TiO_2_. The interest in the Fe^3+^ doping element is due to its cation radius Fe^3+^ (0.65 Å) being similar to that of the Ti^4+^ (0.68 Å) cation; therefore, doping Fe in a TiO_2_ lattice is allowed in principle. In addition, the Fe 3*d* electron configuration similar to that of the Ti atom can offer stability of Fe^3+^ [22].

According to our previous work [26], 1 wt.% Co was demonstrated to be the optimal dopant ratio to develop TiO_2_-based photocatalysts with improved photocatalytic activity toward methyl orange degradation and, thus, the same dopant ratio was used in this work for Co- and Fe-doped TiO_2_. To evaluate the photocatalytic activity of the prepared nanoparticles, carbamazepine was chosen as an organic pollutant. To our knowledge, there is a lack of studies in the literature concerning the photocatalytic degradation of carbamazepine using Fe-doped TiO_2_ nanoparticles as a photocatalyst. Moreover, 1 wt.% Fe-doped TiO_2_ is yet to be used as a photocatalyst in the degradation of carbamazepine. In this work, pure TiO_2_ nanoparticles, Fe-doped TiO_2_ nanoparticles, and Co-doped TiO_2_ nanoparticles were synthesized via wet chemical methods. The structural properties of the prepared samples were analyzed by X-ray diffraction (XRD), Fourier-transform infrared (FTIR) spectroscopy, and Raman spectroscopy, and the optical properties were investigated using UV-visible-light (UV-Vis) diffuse reflectance (DRS) and photoluminescence (PL) spectroscopy. The photocatalytic activity of doped and undoped TiO_2_ nanoparticles was evaluated under both UV-A light and visible-light irradiations, using carbamazepine (CBZ) as an organic pollutant.

## 2. Materials and Methods

### 2.1. Reagents

Titanium(IV) isopropoxide (TTIP, Ti(OCH(CH_3_)_2_)_4_); purity >99.99%)), isopropyl alcohol (purity 99.99%), cobaltous chloride hexahydrate (CoCl_2_, 6H_2_O; purity 99.99%), and iron(III) nitrate nanohydrate (FeN_3_O_9_, 9H_2_O; purity >98%) were purchased from Sigma Aldrich Chemicals (St. Louis, USA). Sodium hydroxide (NaOH; purity 98%) was purchased from Fisher Scientific International Company (Hampton, USA). Carbamazipine (CBZ, C_15_H_12_N_2_O; purity >98%) and acetonitrile (purity >99.5%) were purchased from Honeywell Reidel-de Haën TM (Bucharest, Romania). All chemicals were of analytic grade and used as received without further purifications.

### 2.2. Photocatalyst Preparation

The typical synthesis procedure of TiO_2_ and Co-doped TiO_2_ was adopted from our previous studies, in which pure TiO_2_ was prepared using the sol–gel method [18,25] and Co-doped TiO_2_ synthesis combined the sol-gel and precipitation methods [25]. As for Fe-doped TiO_2_ photocatalyst preparation, the TTIP (Ti(OCH(CH_3_)_2_)_4_) was slowly added dropwise to isopropyl alcohol at room temperature with a molar ratio of 25/1 isopropanol/TTIP, under continuous magnetic stirring; then, a certain amount of H_2_O (molar ratio of H_2_O/TTIP of 100/1) was added dropwise, and a white solution was obtained. Simultaneously, the required mass of the dopant precursor (Fe(NO_3_)_3_) to obtain 1 wt.% Fe in the final product was dissolved in distilled water under constant stirring. We proceeded by adding an aqueous solution of NaOH dropwise at 70 °C. After 90 min under the same conditions of heating (70 °C) and continuous stirring, the obtained solution was added dropwise to the white solution of TiO_2_, prepared as described previously. After 120 min of continuous stirring, the well-mixed solution was filtered and washed with distilled water and dried for 12 h in an oven at 100 °C. The obtained products were ground and annealed at 500 °C in air for 3 h.

### 2.3. Characterization

Powder X-ray diffraction measurement was carried out using an X’PERT MPD_PRO diffractometer (Malvern Panalytical Ltd, Malvern, United Kingdom) with Cu Kα radiation at 45 kV and 40 mA (λ = 1.5406 Å). Fourier-transform infrared (FTIR) spectra of the samples were recorded from 400–4000 cm^−1^ using an FTIR spectrometer type JASCO 4100 (Jasco International, Tokyo, Japan) and the KBr pellet method. The UV-Vis diffuse reflectance spectroscopy measurements were made on a JASCO V-570 spectrophotometer (Jasco International, Tokyo, Japan) equipped with a Labsphere DRA-CA-30I integration sphere, using BaSO_4_ as the reference. Raman spectra were collected at room temperature using a VERTEX 70 apparatus (Bruker Optics, Ettlingen, Germany) with 4 cm^−1^ of spectral resolution. The room-temperature photoluminescence spectra were measured with a Horiba Jovin Fluoromax 4 spectrofluorometer (HORIBA Scientific, Amadora, Portugal).

### 2.4. Photocatalytic Activity Experiments

The photocatalytic experiments were carried out at room temperature (26 ± 2 °C) in a Pyrex cylindric beaker (250 mL) containing 250 mL of a carbamazepine aqueous solution (9 mg/L) and 125 mg of photocatalyst. The photoreactor system was directly exposed to the light source in open-air conditions. After the establishment of the adsorption/desorption equilibrium between CBZ molecules and photocatalyst nanoparticles (60 min in the dark), the suspension was positioned at about 12 cm below the light source (UV-A or visible-light irradiations). In this work, to produce UV-A and visible-light irradiations, a low-pressure lamp (40 W, model Vilber, VL-340.BL, Eberhardzell, Germany) emitting UV radiation at 365 nm (light intensity ≈ 413 mW/cm^2^) and a commercial Feit White Compact Fluorescent lamp (23 W, cool daylight, 6500 K, 1311 Lumens, Mainhouse Electronic Co., Ltd, Xiamen, China) were used, respectively. During the reaction, samples (3 mL) were taken from the suspension and filtered through a 0.45-µm Millipore filter and then analyzed using a reverse-phase (RP) HPLC-diode array detector (DAD) chromatographic system. This system consisted of a Thermo Scientific SpectraSystem P1000 pump, a Rheodyne manual injection valve, a Hypersil Gold column (4.6 mm × 250 mm, 5 µm), and a Thermo Finnigan UV6000 LP diode array detector. The mobile phase consisted of deionized water and acetonitrile (*v*/*v* 40:60) at a flow rate of 1.0 mL∙min^−1^. The volume injected was 20 µL.

## 3. Results and Discussion 

### 3.1. XRD Analysis

Figure 1 presents the XRD patterns for pure TiO_2_ and doped TiO_2_ nanoparticles. These samples only exhibited patterns assigned to the TiO_2_ anatase phase (JCPDS file card No. 21-1272). No signal from the crystalline phase containing metal or metal oxide of the doping elements could be observed, which agrees with previous reports [25,27,28,29]. However, a decrease in the anatase peak intensity was observed for all samples and mainly for Co-doped TiO_2_ NPs, in comparison with undoped TiO_2_ NPs. Furthermore, careful analyses of the main peak (101) of the anatase (inset, Figure 1) indicated a slight shift to the higher angle side for Fe-doped TiO_2_ and mainly for Co-doped TiO_2_ NPs.

Based on this, and the results of a previous X-ray photoelectron spectroscopy (XPS) study that revealed the coexistence of Co^2+^ and Co^3+^ on the surface of Co-doped TiO_2_ NPs [25], and by comparing the cationic radius values of Co^2+^ (0.74 Å), Co^3+^ (0.61 Å), Fe^2+^ (0.76 Å), and Fe^3+^ (0.65 Å) to that of Ti^4+^ (0.68 Å) [25,28,30,31,32], we hypothesize that some of the doping elements were incorporated into the structures of Titania and replaced the titanium ions, which induced a perturbation in anatase crystal structure; as a result, the crystallinity decreased and the peak position shifted. A similar behavior was reported in the literature [15,28,32,33]. The average crystallite size of the samples was estimated from the Full width at half maximum (FWHM) of the prominent peak (101) of anatase, using the Debye-Scherer method [34,35]. The obtained results in Table 1 indicate that the average crystallite size (D) changed after doping TiO_2_ either by Fe or Co. It was found that the crystallite size (D) diminished only by ΔD = 0.05 nm for Fe-doped TiO_2_ NPs, while it increased by ΔD = 0.33 for Co-doped TiO_2_ NPs.

In agreement with previous studies [14,36,37,38,39], we suggest that the Ti^4+^ cation was substituted by low-radius cations Co^3+^ (0.61 Å) and Fe^3+^ (0.65 Å), causing a decrease in d-spacing and shifting peak positions toward the high angle side. Due to the fact that Fe^3+^ has a cation radius value (0.65 Å) very similar to that of Ti^4+^ (0.68 Å) in comparison to Co^3+^ (0.61 Å), the replacement of Ti^4+^ by Fe^3+^ the in Titania lattice structure presented less impact on the crystallinity, peak position, and crystallite size of the TiO_2_ nanoparticles.

### 3.2. FTIR Studies

The FTIR spectra of the prepared samples are presented in Figure 2. The absorption band at 3420 cm^−1^ was attributed to the stretching vibrations of the O-H group adsorbed onto the surface of the nanoparticles, whereas the peak around 1650 cm^−1^ was attributed to the bending vibration mode for the adsorbed water molecules [26,39]. The Fe-doped TiO_2_ spectrum presented a weak absorption band at around 2340 cm^−1^, belonging to the stretching mode of CO_2_ molecules [39]. The band around 1398 cm^−1^ observed on the spectrum of pure TiO_2_ belonged to the bending vibrations of the C-H bond [28,39,40,41]. The low-frequency broad band located in the range 400–900 cm^−1^ corresponded to the Ti-O-Ti vibrational mode [39,42]. Interestingly, a shift of this band position to a lower wave number was observed for both Fe- and Co-doped TiO_2_ NPs, indicating the existence of structure defects [32,43]. In accordance with the XRD results, we think that this band shift was due to the formation of Fe-O and Co-O bonds, which occurred following the substitution of Ti^4+^ with Fe^3+^ and Co^3+^ within the TiO_2_ lattice. This sort of frequency shifting was also presented by other researchers [32,38,43].

### 3.3. Raman Studies

Raman spectroscopy is one of the most efficient analysis techniques to investigate the structural properties of materials. The changes in Raman spectra are related to non-stoichiometry, structure defects, phase changes, and bond modifications [18,32,40,44,45]. Figure 3 shows the Raman spectra of pure TiO_2_, Co-doped TiO_2_, and Fe-doped TiO_2_. All samples exhibited the six Raman active modes, E_g_ (145 cm^−1^), E_g_ (197 cm^−1^), B_1g_ (397 cm^−1^), A_1g_ + B_1g_ (516 cm^−1^), and E_g_ (640 cm^−1^), characteristic of the anatase phase of TiO_2_ [25,27,40,46,47]. No Raman peak from the cobalt or iron was detected, which indicated that the analyzed materials consisted of pure anatase phase, reconfirming the obtained XRD results. Nevertheless, the Raman peak position of E_g_ mode at 145 cm^−1^ was slightly shifted toward a longer wave number, accompanied by a slight decrease in the intensity (inset, Figure 3). A similar behavior of Raman mode signals after doping TiO_2_ NPs with Co or Fe was elsewhere reported [14,29,48], and it is considered as a sign of structure defect existence, which resulted in the present study from the substitution of Ti^4+^ by Co^3+^ and Fe^3+^ within the lattice host. These Raman results agree with the literature and confirm those obtained by XRD and FTIR.

### 3.4. Optical Absorption Studies

The optical properties of doped and undoped TiO_2_ nanoparticles were explored using UV-Vis absorption spectroscopy analyses at room temperature. The recorded absorption spectra are shown in Figure 4a. The absorbance can vary depending upon some factors like particle size, oxygen deficiency, defects in material prepared, etc. [40,49,50]. It is clearly observed on the spectra that the absorption of doped and undoped TiO_2_ NPs was more in the UV region and less in the visible region. More importantly, Figure 4a shows a blue shift of the absorption edge for Fe-doped TiO_2_ and a red shift for Co-doped TiO_2_ nanoparticles, which indicated that the optical properties of TiO_2_ nanoparticles were affected by doping with Fe and Co.

The band gap energy of the prepared NPs was estimated using Tauc’s formula [39,40,51,52].(αhν)^2^ = A (hν − E_g_),(1)where α is the absorbance, and hν is the photon energy. The band gap energy was obtained by extrapolating the linear region of the plot (αhν)^2^ vs. (hν) to intersect the photon energy axis (Figure 4b). The estimated optical band gap (E_g_) value for undoped TiO_2_ was ~3.12 eV, comparable to the value reported in our previous paper (E_g_) of ~3.11 eV [26]. The red shift which occurred for Co-doped TiO_2_ sample was evidenced by its corresponding optical band gap value (E_g_) of ~3.05 eV. These results indicate that cobalt doping helped to reduce the distance between the conduction band and valence band of TiO_2_, which could be favorable for photocatalytic reactions [17,23,53].

The Fe-doped TiO_2_ band gap energy was (E_g_) ~3.32 eV, confirming the blue shift observed on the absorption edge spectra. Similar results were found by many researchers for other doping elements [40,49,54,55,56], and they reported this blue shift to be accompanied by a decrease in the crystallite size. Indeed, in the present case, the previously discussed XRD results revealed a shrinkage in the crystallite size for Fe-doped TiO_2_ nanoparticles. Thus, without neglecting other possible reasons for such a blue shift, such as the Burstein–Moss effect [49,55,56,57,58] or the crystal disorder resulting from the substitution of Ti^4+^ by Fe^3+^ [59,60], and in harmony with the XRD results and the literature, we think that the observed blue shift can be attributed to the well-known quantum-size effect [40,57,61,62,63,64,65].

### 3.5. Photoluminescence Studies

The photoluminescence technique is useful to study the separation and recombination of excited electrons and holes [30,66]. Therefore, all samples were characterized by PL. In this analysis, the excitation of all samples was done at 285 nm at room temperature, and the emission spectra were scanned between wavelengths of 325 and 500 nm; the results are shown in Figure 5. The undoped TiO_2_ PL spectrum shows a strong characteristic UV emission peak centered at 357 nm, which resulted from the band-to-band recombination process of electrons and holes [67,68]. This emission peak red-shifted to 362 and 363 nm in both Fe-doped and Co-doped TiO_2_ sample spectra, respectively, accompanied by an important decrease in intensity. Additional emission peaks centered on 465 nm and 466 nm were observed for Co-doped TiO_2_ and Fe-doped TiO_2_ nanoparticles, respectively, probably arising from oxygen vacancies trapping electrons [67,69]. These charge carriers are generally trapped by oxygen vacancies and surface hydroxyl groups, which contribute to their visible luminescence [67,69].

These observations evidenced that the optical properties of TiO_2_ nanoparticles were significantly affected, and it was suggested that doping TiO_2_ with Fe or Co may significantly suppress the electron-hole pair recombination [19,30,38,70,71], which can also be favorable for photocatalytic reactions [72,73].

### 3.6. Photocatalytic Performance

Based on the structural and optical properties of the prepared nanomaterials, it was expected that they would present high photocatalytic activity in comparison to pristine TiO_2_. To confirm this, experiments were carried out involving the photodegradation of CBZ under UV-A and visible-light irradiations for a period of 240 min. The degradation efficiency due to direct photolysis was also measured under the same experimental conditions as used for photocatalysis (with catalyst). As shown in Figure 6a and Figure 7a the carbamazepine percentage removal via the photolysis process under UV-A and visible light achieved only 9.90% and 3.6% after 240 min of illumination. These results indicate that carbamazepine is resistant to photolysis degradation under UV-A and visible light, and that the simultaneous presence of photocatalyst and light irradiation is necessary for the photocatalytic reaction.

#### 3.6.1. UV-A Light Irradiation

Figure 6 shows the photocatalytic activity test results obtained for pure TiO_2_, Fe-doped TiO_2_, and Co-doped TiO_2_ under UV-A light irradiation. As it can be seen from Figure 6a and Table 2 that the performance of degradation of CBZ reached 96.9%, 34.21%, and 70.06% for 1 wt.% Fe-doped TiO_2_, 1 wt.% Co-doped TiO_2_, and pure TiO_2_ photocatalysts, respectively, after 240 min of UV illumination. These results indicate that the photocatalytic performance of TiO_2_ NPs was significantly improved after doping with Fe, as predicted by the PL results, and unexpectedly decreased when doping TiO_2_ with Co.

The reaction kinetics was further investigated using the Langmuir–Hinshelwood kinetic model, expressed by Equation (2) [57,74,75].ln[C/C_o_] = −k_app_ × t,(2)where C_o_, C, t, and k_app_ represent the initial and time-varying concentrations of CBZ, irradiation time, and apparent kinetic constant, respectively. The plot of −ln(C/C_o_) versus irradiation time (min) yielded a linear relationship as shown in Figure 6b. The correlation coefficients (R^2^) were greater than 0.9 (not presented), and the model fitted well with the experimental data. The apparent first-order rate constants (k_app_) were estimated from the slope of this linear plot, and they were found to be 0.01297 min^−1^, 0.00203 min^−1^, and 0.0067 min^−1^ for 1 wt.% Fe-doped TiO_2_, 1 wt.% Co-doped TiO_2_, and undoped TiO_2_ photocatalysts. The obtained k_app_ values indicated that the kinetics of carbamazepine photodegradation on Fe-doped TiO_2_ was two times faster than that of pure TiO_2_ and six times faster than that of Co-doped TiO_2_.

#### 3.6.2. Visible-Light Irradiation

Figure 7 shows the photocatalytic activity test results obtained under visible-light irradiation for pure TiO_2_, Fe-doped TiO_2_, and Co-doped TiO_2_. The obtained results summarized in Table 2 indicate that 12.54%, 10.77%, and 7.88% from the initial concentration of CBZ was eliminated under visible-light irradiation in the presence of 1 wt.% Fe-doped TiO_2_, 1 wt.% Co-doped TiO_2_, and undoped TiO_2_ photocatalysts, respectively. The corresponding photo degradation kinetics was also investigated using the Langmuir–Hinshelwood kinetic model Figure 7b. The apparent first-order rate constants (k_app_) were 0.00115 min^−1^, 0.00156 min^−1^, and (8.64 × 10^−4^ min^−1^) for Fe-doped TiO_2_, Co-doped TiO_2_, and undoped TiO_2_, respectively. The calculated k_app_ values reflect that the reaction kinetic of carbamazepine photodegradation over TiO_2_ nanoparticles was ameliorated after doping with either Co or Fe.

It is well known that the photocatalytic activity of photocatalysts is affected by a wide variety of factors such as the recombination of photogenerated charge carriers, the crystalline structure, the morphology, and the particle size of the photocatalyst [74,75]. In this work, in a clear way, the photocatalytic experiments results revealed a low CBZ conversion by pristine TiO_2_ under visible-light irradiation, which could be essentially due to its low absorbance of wavelengths above 397 nm (E_g_ = 3.12 eV). Therefore, the electron/hole (e−/h+) pairs, which play an essential role in photocatalytic reactions, cannot be photogenerated under visible-light irradiation. The results show also that, despite the low recombination rate of electrons and holes presented by the PL analyses and the band gap narrowing of Co-doped TiO_2_ nanoparticles, its photocatalytic activity under UV-A light compared to pure TiO_2_ remained below expectations. However, Co doping into the TiO_2_ nanoparticle lattice structure enhanced the photocatalytic performance under visible-light irradiation, via the formation of intermediate energy levels between conduction and valence bands that served as trapping centers for the photogenerated electrons.

The photocatalytic reaction experiment under both UV-A light and visible-light irradiations revealed an important improvement in the photocatalytic activity of TiO_2_ NPs after doping with Fe, which we believe might be ascribed to the high separation rate of electron-hole pairs. These results are in good agreement with those of the PL analyses and those reported in the literature [29,31,39,72]. Luu et al. [29], Asiltürk et al. [31], and Majeed Khan et al. [39] used Fe-doped TiO_2_ for the degradation of *p*-xylene, Malachite Green dye, and methylene blue, respectively. They reported that doping TiO_2_ nanoparticles with Fe^3+^ ion improved their photocatalytic activity. Lin et al. [72] found that 3.4 wt.% Fe-doped TiO_2_ synthesized by hydrothermal deposition showed an improved photocatalytic activity toward carbamazepine degradation as a result of enhancing the separation rate of photo-induced charge carriers by incorporating Fe in the TiO_2_ lattice structure.

## 4. Conclusion

To conclude, bare TiO_2_ nanoparticles, and 1 wt.% Co-doped and 1 wt.% Fe-doped TiO_2_ photocatalysts were successfully prepared via wet chemical methods. Both XRD and Raman results suggested that anatase was the only phase in the samples. Smaller crystallites were found in the Fe-doped TiO_2_ sample. The diffuse reflectance spectra of the samples showed a red shift of the absorbance edge for 1 wt.% Co-doped TiO_2_, and a blue shift was observed for the 1 wt.% Fe-doped TiO_2_ photocatalyst. From the PL plots, reduced intensity emission was observed in Co- and Fe-doped TiO_2_ nanoparticles, which implies an improvement in charge separation efficiency. Photocatalytic experiments revealed that the 1 wt.% Fe-doped TiO_2_ photocatalyst exhibited a stable and remarkably enhanced photocatalytic activity, when compared to bare TiO_2_ and 1 wt.% Co-doped TiO_2_ photocatalysts, toward carbamazepine degradation under UV-A irradiation (96.79% degradation) and under visible-light irradiation (12.54% degradation). The enhancement in photocatalytic efficiency could be due to the reduction of electron/hole recombination.

## Figures and Tables

**Figure 1 materials-12-03874-f001:**
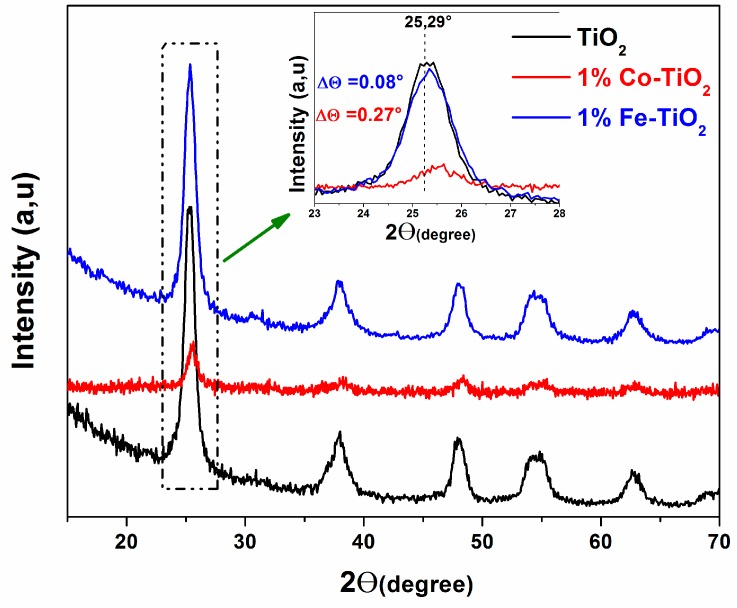
X-ray diffraction (XRD) patterns of pure TiO_2_, and 1 wt.% Co-doped and 1 wt.% Fe-doped TiO_2_ nanoparticles (NPs). The inset shows the peak (101) of anatase.

**Figure 2 materials-12-03874-f002:**
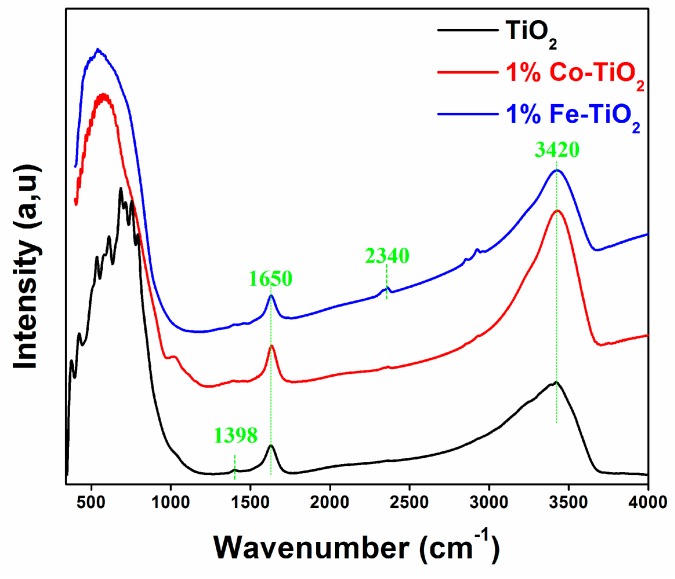
Fourier-transform infrared (FTIR) spectra of pure TiO_2_, and 1 wt.% Co-doped and 1 wt.% Fe-doped TiO_2_ NPs.

**Figure 3 materials-12-03874-f003:**
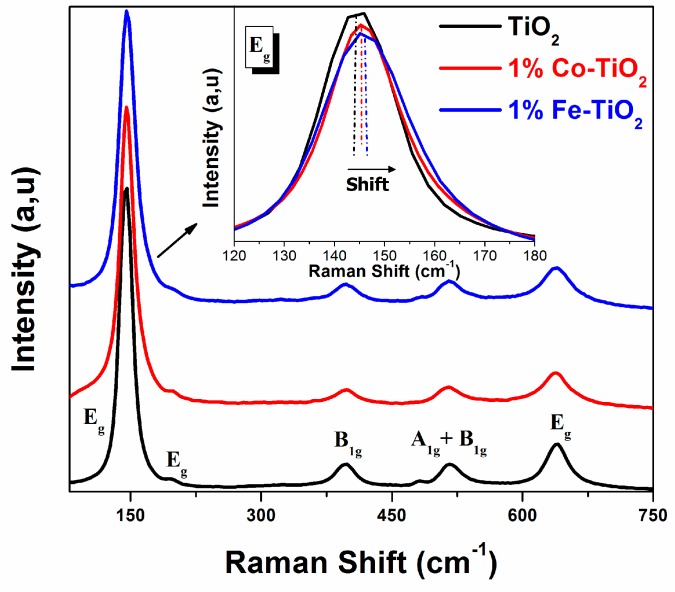
Raman spectra of pure TiO_2_, and 1 wt.% Co-doped and 1 wt.% Fe-doped TiO_2_ NPs.

**Figure 4 materials-12-03874-f004:**
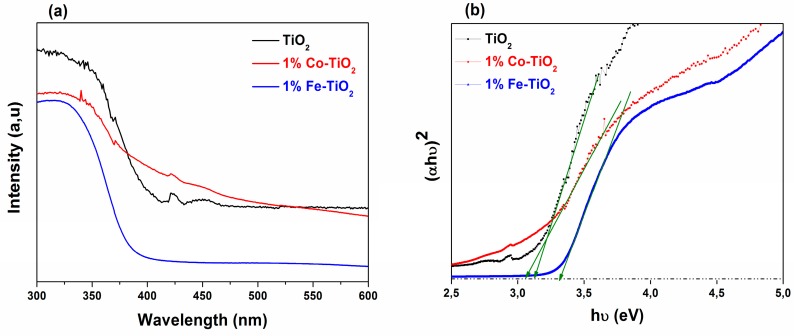
(**a**) Ultraviolet-visible light (UV-Vis) absorbance spectra of pure TiO_2_, and 1 wt.% Co-doped and 1 wt.% Fe-doped TiO_2_ NPs. (**b**) Tauc’s plot.

**Figure 5 materials-12-03874-f005:**
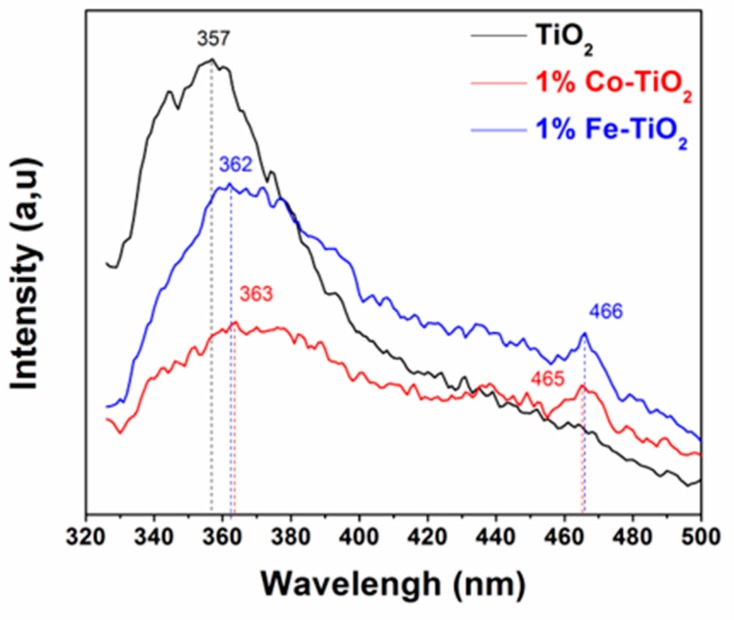
Photoluminescence spectra of pure TiO_2_, and 1 wt.% Co-doped and 1 wt.% Fe-doped TiO_2_ nanoparticles.

**Figure 6 materials-12-03874-f006:**
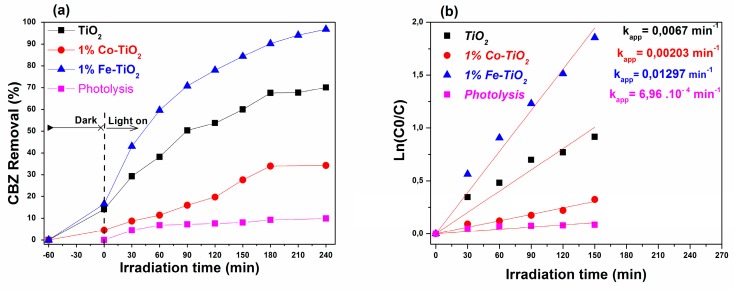
(**a**) Degradation rate of carbamazepine over pure TiO_2_, and 1 wt.% Co-doped and 1 wt.% Fe-doped TiO_2_ under UV-A light irradiation. (**b**) The reaction kinetics data.

**Figure 7 materials-12-03874-f007:**
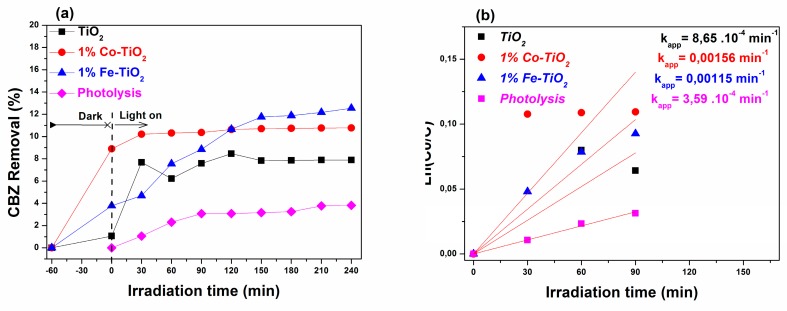
(**a**) Degradation rate of carbamazepine over pure TiO_2_, and 1 wt.% Co-doped and 1 wt.% Fe-doped TiO_2_ under visible-light irradiation. (**b**) The reaction kinetics data.

**Table 1 materials-12-03874-t001:** The peak (101) position, d_hkl_, and crystallite size of pure and doped TiO_2_ nanoparticles.

Samples	Peak (101) Position 2θ (°)	d_hkl_	Average Crystallite Size (nm)
TiO_2_	25.29	0.98	7.45
1 wt.% Fe-TiO_2_	25.37	0.74	7.40
1 wt.% Co-TiO_2_	25.56	0.33	7.78

**Table 2 materials-12-03874-t002:** Band gap energy of TiO_2_-based nanomaterials and the photocatalytic degradation data of carbamazepine (CBZ) under ultraviolet (UV-A) and visible-light irradiation.

Samples	E_gap_	UV-A Light			Visible Light		
		MO Removal (%)	Rate Constant, k (min^−1^)	*R* ^2^	MO Removal (%)	Rate Constant, k (min^−1^)	*R* ^2^
TiO_2_	3.12	70.06	0.0067	0.97	7.88	8.64 × 10^−4^	0.84
1 wt.% Fe-TiO_2_	3.32	96.9	0.01297	0.99	12.54	0.00115	0.97
1 wt.% Co-TiO_2_	3.05	34.21	0.00203	0.99	10.77	0.00156	0.81

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
