# Peer review of "Synthesis of Fe- and Co-Doped TiO_2_ with Improved Photocatalytic Activity Under Visible Irradiation Toward Carbamazepine Degradation"

_materials, 2019, doi:10.3390/ma12233874_

Round 1

Reviewer 1 Report

Manuscript ID: materials-636624

Title: Synthesis of Fe-Doped TiO2 and Co-Doped TiO2 with Improved Photocatalytic Activity Towards Carbamazepine Degradation

The manuscript presents the synthesis and characterization of pure TiO2, Fe-doped and Co-doped TiO2, related to their photocatalytic activity for carbamazepine (CBZ) degradation. TiO2 has been typically synthesized by sol-gel method. Fe- and Co- doping has been performed by sol-gel and precipitation method. XRD, FTIR, RAMAN and UV-VIS absorption analyses were performed to characterized the different materials. The UV and visible light photocatalytic degradation of carbamazepine was carried out at room temperature, using a starting concentration of substrate equal to 9mg/L and 125 mg of photocatalyst, to investigate its photocatalytic performance. Reaction was followed by HPLC analysis. Fe-TiO2 showed the highest CBZ degradation both under UV and visible light. Co-TiO2 showed lower activity than pure TiO2 under UV light, but higher activity under visible light.

Comments for authors

Decision: Rejected

This work presents a series of interesting data, but, in my opinion, the results and their explanation is not sufficient for publication. The most evident fail is that authors reported two different doping of TiO2, i.e. Fe and Co, they found a completely different behavior comparing UV and visible light photocatalytic activity, and they did not explain why from a scientific point of view, using eventually the characterization results. They often refer to other publications, supposing that what they found could be explained in similar way.

Starting from the data they have collected, I suggested to improve both photocatalytic tests and discussion, in order to complete a work which can be suitable for publication.

According to my decision, in the following lines my corrections and questions:

Title:

The tile reports that Fe-doped TiO2 and Co-doped TiO2 has improved photocatalytic activity. However, this is not true for Co-TiO2 under UV light. Why do you select Co for CBZ degradation? What did you expect comparing Fe and Co? Is there a relation between the Me species, substrate degradation, improvement of the photocatalytic activity of TiO2 under a certain type of light?

Abstract:

A brief background, introducing the problem and why this research is important should be added at the beginning. According to my experience in photocatalysis, there is no novelty in Fe-doped or Co-doped TiO2; moreover, the real problem, if present, related to the presence of CBZ and the necessity to find solution to its degradation are not clear.

Introduction:

Line 1: avoid the use of generic phrases not useful for presenting a scientific report.

Line 56: about the non-toxicity of nano-TiO2: the most recent research is moving to new materials, low cost, easy to handle, non-toxic, i.e. not inhalable and easy to recover from liquid system. How to manage nano powders in water? How to manage metal nanoparticles recovery in water?

Lines 60-64: approximate, very general.

Lines 66-67: authors wrote that Co improved the photocatalytic activity, as already reported. Under which light? Why here the results are different?

Why did you decide to synthesized titania by wet chemical methods? Avoid “several spectroscopic techniques” but describe which ones and why

Materials and methods:

The experimental should report enough details in order to reproduce the synthesis. Avoid the use of “certain amount” but use precise quantities. XPS spectroscopy or TEM analyses can improve the materials characterization. About the photocatalytic experiments: The UV light intensity is not reported (according to the distance of the lamp) – the same for visible irradiation source. The problem of temperature increase: using UV light the temperature can increase a lot. In my experience, I usually performed the reaction under UV light using a jacked reactor with cooling system. Did you monitor the temperature during the reaction? Are you completely sure to not have solvent evaporation? 125 mg of catalyst are calculated on the basis of what? Reactions in dark should be performed as the photolysis.

Results and discussion:

Fig 2: add details and number on the figure.

Every time you discuss some values I suggest to avoid imprecise words such as variation, slightly or similar, but use precise quantities and numbers; About the band gap values (Eg): the reported variation is very low, i.e. 3.12 – 3.20 – 3.05 – if you can compare 3.12 to 3.2 referring to pure TiO2, how a decrease to 3.05 can be consider a real decreasing of the band gap? Why Fe-TiO2 shows a band gap of 3.32 eV? Explain why the lower band gap of Fe-titania led to higher activity both in UV and visible light, while Co showed lower activity under UV and improved activity under visible light. Fig 5: my suggestion is to lengthen the lines referring to the peaks until the x axis. Line 222-223: if the presence of Fe or Co suppressed the electron-hole recombination you should have improved activity under UV light in both cases. Moreover, this is a general comment with many references without any accurate explanation.

Photocatalytic performance:

Lines 226-228 are useless. 4 h or 240 min – it is better to maintain the same format Instead of numbers % in the text, a table is clearer. TiO2 requires subscript Lines 267-277: many information without a real explanation of the reason why such catalysts gave these results. Compared to pure TiO2, doping with Fe enhance the activity from 60 % to 95 %. is it profitable from an economical point of view, considering the wide exploitation of commercial and low cost TiO2? Lines 284-290: tentative to give an explanation using other papers. Why the behavior of other catalysts can explain the behavior of these catalysts?

References:

References are correctly reported, however there are several, often used in group, without a real disclosing of them with the respect to this research.

Author Response

Reviewer 1:

We highly appreciate the reviewers’ insightful and helpful comments on our manuscript. We have adopted all the suggestions in our revised manuscript. Below are our point-to-point responses to the reviewer's comments.

Title:

ʻʻThe tile reports that Fe-doped TiO2and Co-doped TiO2has improved photocatalytic activity. However, this is not true for Co-TiO under UV light. ʼʼ

ANSWER:We changed the title as follows:

“Synthesis of Fe and Co-Doped TiO2with Improved Photocatalytic Activity Under Visible Irradiation Towards Carbamazepine Degradation”.

ʻʻ Why do you select Co for CBZ degradation? ʼʼ

ANSWER:It has been reported that doping titanium dioxide with transition metal ions (with different dopant ratio) to be beneficial for narrowing the bandgap and improving the photocatalytic activity of TiO2nanoparticles (J. Environ. Manage, 127 (2013) 142-149; J. Am. Chem. Soc. 135 (2013) 10064−10072; Appl. Catal. A Gen, 495(2015) 131–140; Materials (Basel), 11 (2018) 1946; Materials (Basel), 10 (2017) 631).

In fact, our previous experience (references 25 and 26 of the manuscript) showed that cobalt mono doping of TiO2showed an enhanced photocatalytic activity toward methyl orange degradation under visible light irradiation. In this work, we used this nanoparticle as a reference to which compare the results obtained with Fe-doped TiO2.

ʻʻ What did you expect comparing Fe and Co?Is there a relation between the Me species, substrate degradation, improvement of the photocatalytic activity of TiO under a certain type of light? ʼʼ

ANSWER:The present study aims to examine the structural and optical properties of TiO2doped with Fe, in comparison to Co-doped TiO2and bare TiO2. It is also an examination of the photocatalytic activity of Fe, Co-doped TiO2and pure TiO2nanoparticles toward carbamazepine degradation.

In our previous work (references 25 and 26 of the present manuscript) we investigated the effect of Co, Mn and Ni mono-doping of TiO2, and it was found that Co-doped TiO2showed promising results, concerning the shift of the absorption edge to the visible region and the photocatalytic activity toward methyl orange. That is why in the present study we used the Co as doping element to compare with Fe doping of TiO2. It was also expected to have successful incorporation of Fe3+into TiO2lattice structure due to the similarity of the Fe3+cation radii to that of Ti4+.

Abstract:

ʻʻ A brief background, introducing the problem and why this research is important should be added at the beginning. ʼʼ

ANSWER:We sincerely thank the reviewer for his suggestion. The introduction of the problem and the importance of the work is explained in the introduction. With the modifications made on the revised manuscript (Introduction section), it should be more clear and visible why this research is important.

ʻʻ the real problem, if present, related to the presence of CBZ and the necessity to find solution to its degradation are not clear. ʼʼ

ANSWER:We thank the reviewer for this comment. The answer is presented between lines 36 and 38 of page 1 of the revised manuscript, as follows:

ʻʻMany of these compounds cannot be fully removed by these plants and they are discharged in surface water raising concerns about their potential ecotoxicological effects [1–3]ʼʼ

Also, between lines 42 and 46 of page 2 of the revised manuscript, as follows:

ʻʻAlthough its low concentration in the environment, CBZ is resistant to the biodegradation and to the conventional wastewater treatment processes, and it stays retained in the environment for a long time. Given these facts, CBZ is classified as a persistent organic pollutant [9–11], its accumulation poses a threat to the quality of water resources, and it is suspected to pose toxic effect on the aquatic organisms. [3]ʼʼ

Introduction:

ʻʻ Line 1: avoid the use of generic phrases not useful for presenting a scientific report. ʼʼ

ANSWER:the changes are made in line 34 of page 1, as follow:

ʻʻThe contamination of water systems by pharmaceuticals is recognized as an environmental issueʼʼ

ʻʻ Line 56: about the non-toxicity of nano-TiO2: the most recent research is moving to new materials, low cost, easy to handle, non-toxic, i.e. not inhalable and easy to recover from liquid system.How to manage nanopowders in water? How to manage metal nanoparticlesrecovery in water? ʼʼ

ANSWER:We thank the reviewer for this precious remark. Indeed, there are concerns about the remaining nanomaterials in the water. We can separate them using membranes, but this add to the cost. Under this optic, there is the possibility of immobilized nanoparticles that seem to be attractive solution.

Gnanasekaran, L.; Hemamalini, R.; Saravanan, R.; Ravichandran, K.; Gracia, F.; Gupta, V.K. Intermediate state created by dopant ions (Mn, Co and Zr) into TiO2 nanoparticles for degradation of dyes under visible light. J. Mol. Liq.2016,223, 652–659. Karafas, E.S.; Romanias, M.N.; Stefanopoulos, V.; Binas, V.; Zachopoulos, A.; Kiriakidis, G.; Papagiannakopoulos, P. Effect of metal doped and co-doped TiO2 photocatalysts oriented to degrade indoor/outdoor pollutants for air quality improvement. A kinetic and product study using acetaldehyde as probe molecule. J. Photochem. Photobiol. A Chem.2019, 371, 255–263.

Another interesting approach for water treatment is the use of coagulant for the recovery of the nanoparticles.

Fernández-Ibáñez, P.; Blanco, J.; Malato, S.; De Las Nieves, F.J. Application of the colloidal stability of TiO 2 particles for recovery and reuse in solar photocatalysis. Water Res.2003, 37, 3180–3188. Nurmi, J.T.; Sarathy, V.; Tratnyek, P.G.; Baer, D.R.; Amonette, J.E.; Karkamkar, A. Recovery of iron/iron oxide nanoparticles from solution: Comparison of methods and their effects. Nanoparticle Res.2011, 13, 1937–1952.

ʻʻ Lines 66-67: authors wrote that Co improved the photocatalytic activity, as already reported. Under which light? Why here the results are different? ʼʼ

ANSWER:The light type was specified in the corrected version of the manuscript between lines 62 and 64 of page 2, as follow:

ʻʻwe reported that the mono-doping of TiO2 with Co and its co-doping with (P,Mo), (P,W) or (Si,W) improved the photocatalytic activity of TiO2under visible irradiation, toward methyl orange, by comparison with undoped TiO2[25,26]ʼʼ

In the present study, the obtained results of the photocatalytic activity of Co-doped TiO2NPs under visible light irradiation are coherent with those previously reported for methyl orange degradation. In which Co-doped TiO2showed enhanced photocatalytic activity compared to bare TiO2.

 ʻʻ Why did you decide to synthesized titania by wet chemical methods?

ANSWER:The TiO2synthesis was made by sol-gel method, because it is easy to handle and uses room temperature during preparation, and it can produce fine powder of spherical nanoparticles with uniform size.

 ʻʻ Avoid “several spectroscopic techniques” but describe which ones and why ʼʼ

ANSWER:We appreciate the reviewer`s comment, and corrections were made on the revised manuscript between lines 76 and 78 of page 2, as follows:

ʻʻThe structural properties of the prepared samples were analyzed by XRD, FTIR, Raman spectroscopy, and optical properties were investigated by using UV-vis Diffuse Reflectance (DRS) and photoluminescence (PL) spectroscopy.ʼʼ 

 Materials and methods:

ʻʻ the experimental should report enough details in order to reproduce the synthesis. ʼʼ

ANSWER:More information that would facilitate the reproduction of the synthesis are inserted in the line 95 and 96 of page 3.

 ʻʻThe UV light intensity is not reported the same for visible irradiation source.ʼʼ

ANSWER:We thank the reviewer for this comment. The UV light intensity has been added in the line 121 of page 3. As for the intensity of the visible light it is already written in the line 122 of page 3, which is 1311 Lumens.

“The problem of temperature increase: using UV light the temperature can increase a lot. In my experience, I usually performed the reaction under UV light using a jacked reactor with cooling system. Did you monitor the temperature during the reaction? Are you completely sure to not have solvent evaporation?”

ANSWER:Indeed, carrying out the photocatalytic experiments under UV light irradiation increases the temperature of the reactions, but for the present case we have monitored the temperature of the reaction and it was kept at room temperature by placing the reaction beaker inside an appropriate cooling bath. Therefore, we believe there was no solvent evaporation.

ʻʻ 125 mg of catalyst are calculated on the basis of what? ʼʼ

ANSWER:In this study, we based on our previous works (Catalysis Today 321–322 (2019) 41–51;

ʻʻ Reactions in dark should be performed as the photolysis. ʼʼ

ANSWER:The reactions in the dark were performed to establish the equilibrium of adsorption/desorption of Carbamazepine molecule on the surface of the catalysts. This operation should be carried out in the absence of any irradiation.

Line 117-118 of page 3:

ʻʻAfter the establishment of the adsorption/desorption equilibrium between CBZ molecules and photocatalyst nanoparticles (60 min in the dark)ʼʼ

On the other hand, to verify that the photodegradation of CBZ occurs only in the presence of both the light (UV or visible) and the photocatalyst, black experiments were conducted by placing a CBZ solution under UV or visible irradiation in the absence of photocatalyst.

Line 246-247 page 8:

 ʻʻThe degradation efficiency due to direct photolysis was also measured under the same experimental conditions as photocatalysis (with catalyst) ʼʼ

Results and discussion:

ʻʻ Fig 2: add details and number on the figure. ʼʼ

ANSWER:Some changes were made to Figure 2 in the revised manuscript

ʻʻ Every time you discuss some values I suggest to avoid imprecise words such as variation, slightly or similar, but use precise quantities and numbers ʼʼ

ANSWER:We thank the reviewer for this remark.

ʻʻ About the band gap values (E ): the reported variation is very low, i.e. 3.12 – 3.20 – 3.05 – if you can compare 3.12 to 3.2 referring to pure TiO2, how a decrease to 3.05 can be consider a real decreasing of the band gap? ʼʼ

ANSWER: We thank the reviewer for this correction. We have made the correction on the revised manuscript between lines 215 and 216 of page 7, as follows:

ʻʻThe estimated optical band gap (Eg) value for undoped TiO2was ~ 3.12 eV, which is comparable to the value reported in previous paper (Eg) ~ 3.11 eV [26].ʼʼ

ʻʻ Why Fe-TiO2 shows a band gap of 3.32 eV? ʼʼ

ANSWER:The explanation was suggested in the between line 220 and 223 of page 7,as follow:

ʻʻwithout neglecting other possible reasons of such blue shift, such as Burstein-Moss effect [49,55–58] or the crystal disorder resulted from the substitution of Ti4+by Fe3+[59,60] and in line with the XRD results and the literature, we think that the observed blue shift can be attributed to the well-known quantum-size effect  [40,57,61–65]ʼʼ

ʻʻ Explain why the lower band gap of Fe-titania led to higher activity both in UV and visible light, while Co showed lower activity under UV and improved activity under visible light. ʼʼ

And 

ʻʻ Line 222-223: if the presence of Fe or Co suppressed the electron-hole recombination, you should have improved activity under UV light in both cases. ʼʼ

ANSWER:Indeed, the PL results suggest the possibility of higher photocatalytic activity for Co and Fe-doped TiO2nanoparticles by comparison to undoped TiO2, which is the case for the photocatalytic activity under visible light. We have explained these results in the revised version of the manuscript in Lines 293-306. The reason of the low photocatalytic activity of TiO2under visible light irradiation is its wide band gap.

As for the low photocatalytic, activity under UV light for Co-doped TiO2, (although the low bad gap and the low recombination rate of electron hole observed for Co-doped TiO2) compared to pristine TiO2, we suspect that these results have a relation with thespecific surface of the nanoparticles. We assume that the specific surface area of TiO2NPs decreased when doped with Co leading to low contact of the nanoparticles with the light irradiation and less adsorption sites for the photocatalytic reaction.

ʻʻ Fig 5: my suggestion is to lengthen the lines referring to the peaks until the x-axis ʼʼ

ANSWER:We appreciate the reviewer's suggestion. Figure 5 has been modified on the revised manuscript.

Photocatalytic performance:

ʻʻ Lines 226-228 are useless. 4 h or 240 min – it is better to maintain the same format Instead of numbers % in the text, a table is clearer. TiO2requires subscript Lines

ANSWER:We appreciate the reviewer`s remark, and we have made the required changes on the format.

ʻʻ 267-277: Compared to pure TiO2, doping with Fe enhance the activity from 60 % to 95 %. is it profitable from an economical point of view, considering the wide exploitation of commercial and low cost TiO2? ʼʼ

ANSWER:Considering the low cost and the easy steps of the synthesis of such catalyst and considering the improved photocatalytic activity of the obtained nanomaterials, Fe-doped TiO2photocatalysts could be a good economical alternative of pure TiO2, especially if we use visible light irradiation. However, more studies and exploration of the effect of other dopant ratio are required and more photocatalytic experiments on other pollutants would be necessary.

ʻʻ Lines 284-290:tentative to give an explanation using other papers. Why the behavior of other catalysts can explain the behavior of these catalysts? ʼʼ

ANSWER:We thank the reviewer for this comment. On which we respectfully disagree, the studies cited in the manuscript between lines 308 and 310 page 10 are not for explanation but they are there as a comparison.

The explanation was given between line 304 and 306 of page 10, as follows:

ʻʻThe photocatalytic reaction experiment under both UV-A light and visible light irradiations revealed on an important improvement of the photocatalytic activity of TiO2NPs after doping with Fe, which we believe it might be ascribed to high separation rate of electron-hole pairsʼʼ

Reviewer 2 Report

Comments to the Authors: 

In this manuscript, the authors prepared Co and Fe-doped TiO2 nanoparticle using wet chemical methods. They concluded that 1 wt.% dopant would enhance the charge separation efficiency. Also, doping TiO2 with Fe results in increasing photocatalytic activity, and subsequently, better carbamazepine degradation. 

This manuscript is written well and appealing. I would recommend it for publication in Materials after minor revisions.

“However a decrease of the anatase peaks intensity was observed for all samples and mainly for Co-doped TiO2 NPs” The decrease of the peaks intensity is observed only for Co-doped TiO2, not for all! The authors should explain Figure 1 in more detail and describe why peaks intensity of 1% Co-TiO2 is decreased compared to pure TiO2. 

Figure 3. subset does not clearly show the Raman shift of Eg mode.

More comparison with refs 14,22-24, 29,48 are required to demonstrate what has been done in those refs and what this manuscript adds to the previous works.

The significance of the work is not discussed well. More reasoning is needed to justify the importance of such work. 

Author Response

Reviewer 2:

We thank the reviewer for all the constructive comments, which helped us to improve the quality of our manuscript. We have adopted all the suggestions in our revised manuscript. Below are our point-to-point responses to the reviewer's comments.

This manuscript is written well and appealing. I would recommend it for publication in Materials after minor revisions

ANSWER:We thank the reviewer for these words.

The authors shouldexplain Figure 1 in more detail and describe why peaksintensity of 1% Co-TiO2is decreased compared to pureTiO2.”

ANSWER:This question was addressed as follows, between lines 149 and 152 of page 4:

we hypothesize that the doping elements were incorporated into the structures of titania and replaced the titanium ions, which induced a perturbation in anatase crystal structure and as a result, the crystallinity decreased and the peaks position is shifted, similar behavior has been reported in the literature [15,28,32,33]”

Figure 3. Subset does not clearly show the Raman shift of Eg mode.

ANSWER:We have made some changes in Figure 3 to make more visible the Eg Raman mode shifting.

More comparison with refs 14, 22-24, 29, 48 are required to demonstrate what has been done in those refs and what this manuscript adds to the previous works.”

ANSWER:The comparison was made as follows, between lines 306 and 313 of pages 10 and 11:

“These results are in good agreement with those of the PL analyses and those reported in the literature [29,31,39,72].  Cam Loc Luu et al. [29], Meltem Asiltürk et al. [31] and M.A. Majeed Khan et al. [39] who used the Fe doped TiO2for the degradation of p-xylene, Malachite Green dye and methylene blue, respectively. They reported that doping TiO2nanoparticles with Fe3+ion improved their photocatalytic activity. Lu Lin et al. [72] found that 3.4 wt% Fe-doped TiO2synthesized by hydrothermal deposition showed an improved photocatalytic activity toward carbamazepine degradation as a result of enhancing the separation rate of photo-induced charge carriers by incorporating Fe in TiO2lattice structure.”

“The significance of the work is not discussed well. More reasoning is needed to justify the importance of such work.”

ANSWER:The importance of the work more explained in the revised version of the manuscript follows, between lines 65 and 75 of page 2:

“Therefore, in order to continue our studies about the effect of doping with transition metal ions on the structural, optical and photocatalytic properties of TiO2nanoparticles, the present study is made to examine the effect of Fe doping on TiO2nanoparticles, compared to Co doping and pure TiO2. According to our previous work [26], 1 wt% of Co demonstrated to be the optimal dopant ratio to develop TiO2-based photocatalysts with improved photocatalytic activity toward methyl orange degradation. Therefore, the same dopant ratio was used in this work for Co and Fe-doped TiO2. To evaluate the photocatalytic activity of the prepared nanoparticles, carbamazepine was chosen as organic pollutant. To our knowledge, there is a lack of studies in the literature concerning the photocatalytic degradation of carbamazepine using Fe-doped TiO2nanoparticles as photocatalyst. Moreover, 1wt% Fe-doped TiO2has not yet been used as a photocatalyst in the degradation of carbamazepine.”

Reviewer 3 Report

The manuscript reports the synthesis of pure, cobalt-doped, and iron-doped TiO2 and their application for photocatalytic degradation of carbamazepine (CBZ) which is a prescription drug. Several characterization techniques were used including XRD, FTIR, PL and DRS. The manuscript is well-organized and easy to follow. However, the current version of the manuscript lacks very important information regarding the novelty statement. It is not clear if this is the first study on the photocatalytic degradation of CBZ or not, as this is, for sure, not the first study on the degradation using an AOP (ref. 12 and 13). In the revised manuscript, please carefully elaborate on this. In addition, it is not clear why Co and Fe elements were selected. Both of these elements were previously investigated as doping elements to make the TiO2 photocatalysis visible-light-driven. It is not clear what is the main goal of the manuscript. Is it synthesis a visible-light-driven photocatalysis and use it for an organic pollutant removal?  Is this research designed to specifically remove CBZ with a visible-light-driven photocatalysis, so they had to dope TiO2 with some elements? Please clarify this in the introduction.

Some technical comments:

The PL results show a higher visible-light performance for the doped TiO2 compared to that of the pristine. However, the photocatalytic degradation under visible light irradiation (Fig. 6) implies that the pristine TiO2 has a higher degradation performance compared to that of the Co-doped one. Please explain possible reasons in detail. Anatase was the dominant TiO2 phase. Please compare this finding with those of the literature in which the same synthesis technique (wet chemical method) was implemented.

Author Response

Reviewer 3:

We truly appreciate all the constructive comments and suggestions from the reviewer. We have adopted all the suggestions in our revised manuscript.Below are our point-to-point responses to the reviewer's comments.

ʻʻThe manuscript is well-organized and easy to follow ʼʼ

ANSWER:We thank the reviewer for these words.

ʻʻthe current version of the manuscript lacks very important information regarding the novelty statement. It is not clear if this is the first study on the photocatalytic degradation of CBZ or not ʼʼ

ANSWER:This is not the first attempt to study the photocatalytic degradation of CBZ by a photocatalysis process, which is mentioned in the corrected version in the lines 48-50 of page 2. However, there are still y few studies concerning the degradation of carbamazepine using Fe- and Co- doped TiO2photocatalysts, a point we made clear in the introduction on lines 72-75 of page 2.

Lines 48-50 page 2:

Among the various AOPs processes the heterogeneous photocatalysis is selected as one of the best options for the destruction of many recalcitrant organic pollutants including CBZ [12,13].”

Line 72-75page 2:

 “To our knowledge, there is a lack of studies in the literature concerning the photocatalytic degradation of carbamazepine using Fe-doped TiO2nanoparticles as photocatalyst. Moreover, 1wt% Fe-doped TiO2has not yet been used as a photocatalyst in the degradation of carbamazepine.

ʻʻit is not clear why Co and Fe elements were selectedʼʼ

ANSWER:The answer was provided on line 59-60 of page 2, as follows:

 “Doping with transition metal ions, such as Fe, Co, Mn, Ni ions, is reported to be effective for enhancing the photocatalytic activity of TiO2[22–24].”

More information was also added on lines 65-67 of page 2.

 “in order to continue our studies about the effect of doping with transition metal ions on the structural, optical and photocatalytic properties of TiO2nanoparticles, the present study is made to examine the effect of Fe doping on TiO2nanoparticles, compared to Co doping and pure TiO2.”

ʻʻIt is not clear what the main goal of the manuscript ʼʼ

ANSWER:The answer is in the lines 65-67 of page 2 in the revised manuscript, as follows:

 “Therefore, in order to continue our studies about the effect of doping with transition metal ions on the structural, optical and photocatalytic properties of TiO2nanoparticles, the present study is made to examine the effect of Fe doping on TiO2nanoparticles, compared to Co doping and pure TiO2.”

Some technical comments:

ʻʻ The PL results show a higher visible-light performance for the doped TiO2compared to that of the pristine. However, the photocatalytic degradation under visible light irradiation (Fig. 6) implies that the pristine TiO2has a higher degradation performance compared to that of the Co-doped one. Please explain possible reasons in detail ʼʼ

ANSWER:With all the respect to the reviewer, we think that he meant by his comment the photocatalytic degradation under UV light irradiation and not under visible light irradiation.

Indeed, the PL results suggest the possibility of higher photocatalytic activity for Co and Fe-doped TiO2nanoparticles by comparison to undoped TiO2, which is the case for the photocatalytic activity under visible light. We have explained these results in the revised version of the manuscript in the lines 293-306 of page 10. The reason of the low photocatalytic activity of TiO2under visible light irradiation is its wide band gap.

As for the low photocatalytic activity under UV light for Co-doped TiO2, (although its low band gap and low recombination rate of electron hole), we suspect that these results have a relation with thespecific surface of the nanoparticles. We assume that the specific surface area of TiO2NPs decreased when doped with Co, leading to less adsorption sites for the photocatalytic reaction.

ʻʻ Anatase was the dominant TiO2phase. Please compare this finding with those of the literature in which the same synthesis technique (wet chemical method) was implemented ʼʼ

ANSWER:the comparison is already made between the lines 142 and 144 of page 4, as follow:

 “No signal from crystalline phase containing metal or metal oxide of the doping elements could be observed, which agrees with previous reports [25, 27–29].” 

Round 2

Reviewer 1 Report

Dear authors,

I appreaciate your efforts to improve the paper. I think now it is suitable for publications, being the description and the discussion more complete.

Author Response

Reviewer 1

We thank the reviewer for the comments. We have made minor adjustments in the revised text (text highlighted).

Reviewer 3 Report

The authors addressed many technical issues of the original manuscript; yet a very important point is missing: the novelty.

The authors claim "there is a lack of studies in the literature concerning the photocatalytic degradation of carbamazepine using Fe-doped TiO2 nanoparticles as photocatalyst". This is NOT a good justification for designing this study. If that's the case, why Co was studied?

Later, the authors justify selecting Fe and Co based on the argument "Doping with transition metal ions, such as Fe, Co, Mn, Ni ions, is reported to be effective for enhancing the photocatalytic activity of TiO2 [22–24].". There are many other transition metals that were proven to be as effective (and sometimes more effective) as/than this list. There are many other non-metal single or couple-doped TiO2 which perform well under visible light (N, S, etc.). The reason to pick Fe or Co MUST be clearly justified. For example, a low cost, unique characteristics, etc. could be a justification.

Author Response

Reviewer 3

ʻʻ The authors addressed many technical issues of the original manuscript; yet a very important point is missing: the novelty.

The authors claim "there is a lack of studies in the literature concerning the photocatalytic degradation of carbamazepine using Fe-doped TiO2 nanoparticles as photocatalyst". This is NOT a good justification for designing this study. If that's the case, why Co was studied?ʼʼ

Answer:The novelty in this work is about the Fe-doped TiO2nanoparticles with enhanced photocatalytic activity. As for Co-doped TiO2NPS, In fact, our previous experience (references 25 and 26 of the manuscript) showed that cobalt mono doping of TiO2showed an enhanced photocatalytic activity toward methyl orange degradation under visible light irradiation. In this work, we used this nanoparticle as a reference to which compare the results obtained with Fe-doped TiO2. This was mentioned in the manuscript between line 60 and line 67

"In fact in our previous studies, which include the effect of Mn, Co, and Ni mono-doping on TiO2nanoparticles and the effect of its co-doping with metal and non-metal ions, we reported that the mono-doping of TiO2with Co and its co-doping with (P,Mo), (P,W) or (Si,W) improved the photocatalytic activity of TiO2under visible irradiation, toward methyl orange, by comparison with undoped TiO2[25,26].

 Therefore, in order to continue our studies about the effect of doping with transition metal ions on the structural, optical and photocatalytic properties of TiO2nanoparticles, the present study is made to examine the effect of Fe doping on TiO2nanoparticles, compared to Co doping and pure TiO2. "

ʻʻ Later, the authors justify selecting Fe and Co based on the argument "Doping with transition metal ions, such as Fe, Co, Mn, Ni ions, is reported to be effective for enhancing the photocatalytic activity of TiO2 [22–24].". There are many other transition metals that were proven to be as effective (and sometimes more effective) as/than this list. There are many other non-metal single or couple-doped TiO2 which perform well under visible light (N, S, etc.). The reason to pick Fe or Co MUST be clearly justified. For example, a low cost, unique characteristics, etc. could be a justification.ʼʼ

Answer:In this work, Fe was selected as doping element for the following reasons: the Fe is very cost-effective, also Fe is a typical transition metal element that have the same 3d orbital as Ti atom, and similar (Fe3+) cation radiu to that of Ti4+cation, therefore doping Fe in TiO2 lattice is allowed in principle, and could modifier the structural and optical properties, and could enhance the photocatalytic activity as reported by many authors. Moreover, Fe-doped TiO2is obtained by easy technique, low price, and low energy consummation (low synthesis temperatures)

The interest of choosing Fe as doping element was modified in the revised manuscript between line 68 and line 70 as follow:

ʻʻThe interest of Fe3+doping element is due to its cation radiu similar to that of Ti4+cation, therefore doping Fe in TiO2 lattice is allowed in principle. In addition, the Fe 3d electron configuration similar to Ti atom can offer stability of Fe3+ʼʼ